



# Performance of the IRI-2016 at the Brazilian low-latitude ionosphere over the South America Magnetic Anomaly during solar minimum

Juliano Moro[1,2], Jiyao Xu[1], Clezio Marcos De Nardin[3], Laysa Cristina Araújo Resende[1,3], Régia Pereira Silva[3], Sony Su Chen[3], Giorgio Arlan da Silva Picanço[3], Liu Zhengkuan[1], Hui Li[1], Chunxiao Yan[1], Chi Wang[1], and Nelson Jorge Schuch[2]

[1]State Key Laboratory of Space Weather – NSSC/CAS, Beijing, China
[2]Southern Regional Space Research Center – CRCRS/COCRE/INPE-MCTIC, Santa Maria, Brazil
[3]National Institute for Space Research – INPE/MCTIC, São José dos Campos, Brazil

*Correspondence to*: Juliano Moro (juliano.moro@inpe.br)

**Abstract.** In this work we analyse the ionograms obtained by the recent Digisonde installed in Santa Maria (29.7° S, 53.7° W, dip angle = -37°), Brazil, to calculate the monthly averages of the $F_2$ layer critical frequency ($f_oF_2$), its peak height ($h_mF_2$), and the E-region critical frequency ($f_oE$) acquired during geomagnetically quiet days from September 2017 to August 2018. The monthly averages are compared to the 2016 version of the International Reference Ionosphere (IRI) model predictions in

order to study its performance close to the center of the South America Magnetic Anomaly (SAMA), which is a region particularly important for High Frequency (HF) ground-to-satellite navigation signals. The $f_oF_2$ estimated with the Consultative Committee International Radio (CCIR) and International Union of Radio Science (URSI) options predicts well throughout the year. Whereas, for $h_mF_2$, it is recommended to use the SHU-2015 option instead of the other available options (AMTB2013 and BSE-1979). The IRI-2016 model outputs for $f_oE$ and the observations presented very good agreements.

**1 Introduction**

The growing importance of space technologies through satellites for a large variety of applications such as science, Earth observation, meteorology, communications, security, and defence, puts forward the need to improve our ability of ionospheric modelling. For instance, the drag force on satellites in low-Earth orbit (LEO, generally defined as an orbit below an altitude of approximately 2,000 kilometer) increases when the solar activity is at its greatest over the 11-year solar cycle,

which may cause uncontrolled re-entry and degrade the predictions of satellite positions (Horne et al., 2013). During space weather conditions as defined by Denardini et al. (2016), elevated flux levels of high energetic particles may precipitate in the ionosphere in regions of anomalously weak geomagnetic field strength such as the South America Magnetic Anomaly (SAMA). Besides enhancing the ionization distribution and conductivities (Moro et al., 2013, 2012), the energetic particles create high background counts which render satellite sensors unusable in this region (Schuch et al., 2019; Heirtzler, 2002).

Operators who control satellites in LEO may need to know with a high degree of accuracy when and where to turn satellites





on and off to minimize the risk of detector saturation (Jones et al., 2017). Ionospheric modelling is also important for ground assets since it is essential to predict the ionospheric behavior for successful radio communication. Since drastic ionospheric variations can affect the performance of radio-based systems, such prediction may identify the periods, the path regions and the sections of high-frequency bands that will allow or disrupt the use of the radio transmissions (Ezquer et al., 2008).

One of the most widely used ionospheric models is the International Reference Ionosphere (IRI), which became the official International Standardization Organization (ISO) standard for the ionosphere since April 2014 (Bilitza et al., 2017). IRI is a joint project of the Committee on Space Research (COSPAR) and International Union of Space Science (URSI). It is derived from ionospheric observations collected by ground and in situ measurements as the worldwide network of ionosondes, incoherent scatter radars, several compilations of rocket measurements, and satellite data. The model describes monthly

averages of the electron density, electron and ion temperature, total electron content (TEC), and ion composition as a function of height, location, and local time. Several major milestone editions of IRI were released by the IRI Working Group since the 1970s in order to constantly revising the model to remain it up to date and accurate as possible (Rawer et al., 1978a, 1978b, 1981; Bilitza, 1990, 2001; Bilitza and Rawer, 1996; Bilitza and Reinisch, 2008; Bilitza et al., 2014, 2017). The latest version is known as IRI-2016 and has important improvements over the 2012 and 2007 versions (IRI-2012 and IRI-2007,

respectively). The most important update is the inclusion of two new model options for the $F_2$ layer peak height, $h_mF_2$. These two options allow the users to model the $h_mF_2$ directly and no longer depend on the propagation factor $M(3000)F_2$ described by Bilitza et al. (1979). Besides, IRI-2016 has a better representation of topside ion densities during very low and high solar activities. The details about the IRI model are available in the following homepage: http://irimodel.org/.

Among several parameters, IRI can predict the $F_2$ layer critical frequency ($f_oF_2$), $h_mF_2$, and the E-region critical frequency

($f_oE$) for a given time and location. The correct understanding of these parameters is particularly important for space technologies. The critical frequencies are two key parameters when calculating the electron densities of the ionosphere at $F_2$ ($N_mF_2$) and E heights. Moreover, $f_oF_2$ is related to the maximum usable frequency for the radio waves reflection and TEC that is significant for the phase delay of High Frequency (HF) ground-to-satellite navigation signals (Fuller-Rowell et al., 2000). On the other hand, $h_mF_2$ receives much of the attention since it gives the highest stratification of the upper ionosphere.

In the literature, several papers have reported many comparative studies around the globe between the ionospheric parameters measured by ionosondes and different versions of the IRI to study its performance. In South America, Ezquer et al. (2008) analysed $N_mF_2$ over Tucumán (26.9° S, 66.4° W, dip angle = -26°), Argentina, during the low and high solar activity years 1965 and 1970, respectively, and the moderate solar activity years 1967 and 1972. Bertoni et al. (2006) used $f_oF_2$ and $h_mF_2$ measured by two digital ionosondes installed at two Brazilian low-latitude stations in July 2003, October 2003,

January 2004, and April 2004. They compared the data collected in Palmas (10.1°S, 48.2°W, dip angle = -12°) and São José dos Campos (23.2° S, 45.8° W, dip angle = -33°) with the IRI-2001 predictions. Batista and Abdu (2004) compared the parameters $f_oF_2$, $h_mF_2$, and $B0$ measured by two digital ionosondes over São Luís (2.6° S, 44.2° W, dip angle = -4.3°, magnetic equator), and Cachoeira Paulista (22.7° S, 45° W, dip angle = -33.5°, close to the southern crest of the Equatorial Ionization Anomaly - EIA) with the IRI-2007 for high and low solar activity periods. Moro et al. (2016) tested the influence



of IRI-2007 in deriving the conductivity profiles and electric files in the Brazilian equatorial region. In Africa, Oyekola and Fagundes (2012) compared $f_oF_2$, $h_mF_2$ and propagation factor ($M3000F2$) recorded near dip-equator Ouagadougou, Burkina Faso (12° N, 1.8° W; dip angle = 2.9°) with IRI-2007 during low (1987) and high (1990) solar activity, and undisturbed conditions for four different seasons. In Europe, Maltseva and Poltavsky (2009) investigated several aspects of the IRI accuracy and efficiency for long term prediction of the $f_oF_2$ and the maximum usable frequencies (MUF) using the storm-time correction option, TEC, and the maximum observable frequency (MOF) for the year 2005. In China, Zhao et al. (2017) used $h_mF_2$ data derived by ionosondes at Mohe, Beijing, Wuhan and Sanya ranging from year 2007 to 2016 to assess the performance of the three options for the IRI- $h_mF_2$, while Liu et al. (2019) used $f_oF_2$ measured over four stations in China (covering from 49.4º N to 23.2º N) from January 2008 to October 2016 to test IRI-$f_oF_2$. The aforementioned studies show that the ionospheric parameters predicted by the IRI model differed from the ionosonde data at a different location. Generally, IRI overestimates the ionospheric parameters at the magnetic equator and underestimate at EIA crests.

The aim of this work is to use the critical frequencies $f_oF_2$ and $f_oE$ and the height $h_mF_2$ measured by a recent Digisonde Portable Sounder 4D (DPS-4D) installed in Santa Maria (29º S, 54º W, dip anlge = -37°), Brazil, to test the performance of the IRI-2016 in the low-latitude ionosphere situated close to the center of the SAMA. The Santa Maria Digisonde (SMK29) is supported by the Space Weather Monitoring Meridian Project of China (Wang, 2010), the Brazilian Study and Monitoring of Space Weather (Embrace) Program from the Brazilian National Institute for Space Research (INPE/MCTIC), and Federal University of Santa Maria (UFSM). Notice that there are very few ionospheric sounders operating in real-time in the low- and mid-latitudes in South America, and SMK29 fills a gap of ionospheric sounding between Cachoeira Paulista station and Port Stanley station (51.6º S, 57.9º W, dip = -49.8º), Argentina. Therefore, validate the IRI-2016 in a region under the influence of the SAMA is particularly important for HF communication and radio-based space systems as described before, besides contributing with IRI Working Group evaluating the goodness of the model in the low latitude Brazilian region.

## 2 Observed Data, Modelling, and Method of Analysis

The SMK29 is set to transmit radio waves continuously into the ionosphere from 1 MHz and increases the frequency up to 20 MHz with the sweep rate of 25 kHz for each round. The train of echoes to form an ionogram is transmitted/received with a 5 minutes temporal resolution. All recorded ionograms are initially auto-scaled by the Automatic Real-Time Ionogram Scaler with True Height (ARTIST). Then, the observed $f_oF_2$, $f_oE$, and $h_mF_2$ parameters are deduced from manually scaled ionograms with help of the Digisonde Ionogram Data Visualization/Editing Tool (SAO-Explorer) developed by the Center for Atmospheric Research, University of Lowell Massachusetts.

Data used in this work were collected in geomagnetic quiet days ($\sum$ Kp $\leq$ 24, where $\sum$ Kp is the sum of the eight 3-h Kp indices for the day) from September 2017 to August 2018. The period is characterized by a very low level of solar and magnetic activity. The 27-day averaged values of the $F_{10.7}$, the sunspot numbers, and the numbers of monthly quiet data used in this work are shown in Table 1. The average of the solar emission at a wavelength of 10.7 cm from September 2017 to



August 2018 is only $(71.6\pm3.5) \times 10^{-22}$ Wm$^{-2}$ Hz$^{-1}$, and the sunspot number range from 1 to 18, characterizing the low solar activity period.

Monthly average values of the observed $f_oF_2$, $f_oE$, and $h_mF_2$ parameters are calculated from the daily hourly values. The IRI-
2016 predictions of $f_oF_2$, $f_oE$, and $h_mF_2$ are computed for the same geophysical conditions to compare with the observational data and to evaluate the discrepancies and goodness of the model. The Relative Deviation (*RD*) of the predicted values concerning to the observed values for modelling the $f_oF_2$ using the Consultative Committee on International Radio (CCIR) coefficient (CCIR, 1967) had been computed through the Eq. (1).

$$f_oF_{2\ CCIR-RD} = \left(\frac{f_oF_{2\ CCIR} - f_oF_{2\ Observed}}{f_oF_{2\ Observed}}\right) \times 100\ \%. \qquad (1)$$

The term $f_oF_{2\ CCIR}$ stands for the monthly average of the $f_oF_2$ modelled by the CCIR sub-routine, while the term $f_oF_{2\ Observed}$ is the monthly average of $f_oF_2$ measured by the SMK29. Besides the comparison between the observed $f_oF_2$ with CCIR, the sub-routine URSI (Rush et al., 1989) is also tested and, therefore, Eq. (1) is also used considering $f_oF_{2\ URSI}$ instead of $f_oF_{2\ CCIR}$.
The $f_oF_2$ storm model (Araujo-Pradere et al., 2002) was turned off in the IRI-2016 options since we are interested in the quiet time conditions. The *RD* is also evaluated for $h_mF_2$ and $f_oE$ using Eq. (1). For $h_mF_2$, the comparison is made considering the currently three options for determining IRI-$h_mF_2$: AMTB2013 (Altadill et al., 2013), SHU-2015 (Shubin, 2015), and BSE-1979 (Bilitza et al., 1979), called AMTB, SHU and BSE, respectively, hereafter. The AMTB model is based on ionospheric data deduced of ionograms recorded by 26 Digisondes embracing latitudes from 65ºN to 52ºS and the longitude sector from
120ºW to 170ºE. The data cover different levels of solar activity from 1998 to 2006. The spherical harmonic technique was applied in AMTB to model the quiet pattern of the $h_mF_2$ at a global scale. The SHU model is based on the ionospheric radio-occultation data collected by CHAMP (from 2001 to 2008), GRACE (from 2007 to 2011) and COSMIC (from 2006 to 2012) satellite missions and ionospheric sounding data collected by 62 Digisondes from 1987 to 2012. SHU uses the spherical harmonics decomposition to model $h_mF_2$. Finally, the older BSE uses the correlation between $h_mF_2$ and propagation factor
$M(3000)F_2$ which in turn is defined by the ratio between the highest frequency that, refracted by the ionosphere, can be detected at a distance of 3,000 km ($M(3000)$) and $f_oF_2$. At last, the $f_oE$ comparison is made using IRI-$f_oE$ developed by Kouris and Muggleton (1973a, 1973b) for CCIR (1973) with a modified zenith angle introduced by Rawer and Bilitza (1990) to improve the nighttime variations. Finally, to evaluate the performance of IRI-2016, a correlation analysis is performed between the modelled parameters and the observational data.
In some cases, the results are discussed considering the seasonal differences between the observed and modelled parameters. Each season is composed by three months as follow: December solstice (November, December, and January), March equinox (February, March, and April), June solstice (May, June, and July), and September equinox (August, September, and October). The local time (LT) in Santa Maria is defined as the universal time (UT) less three hours (LT = UT – 3 h). Finally, since the focus of this work is to analyse the IRI-2016 predictions, the reader can find the complete study about the





variabilities of the $F_2$ and E layers parameters over Santa Maria during the period analysed in the recent work by Moro et al.
        (2019).

## 3 Results

### 3.1 Performance of IRI-$f_oF_2$

The contour plots of the monthly averaged $f_oF_2$ (in MHz) observed and modelled by CCIR and URSI sub-routines, the *RD*

(in percent) versus universal time (UT, vertical axis) and month (from September 2017 to August 2018, horizontal axis) are
        shown in Fig. 1. The $f_oF_2$ values are represented by the color-coded bar on the right-hand side of the upper panels and vary
        from 1 MHz to 12 MHz. The *RD* in the lower panels ranges by ± 50 %.
        During the whole year analysed the sunrise time varied from 9:15 UT to 10:30 UT, while the sunset took place between
        20:43 UT and 23:30 UT over Santa Maria. The observed averaged $f_oF_2$ values depicted in Fig. 1(a) shows a diurnal variation

pattern with highest values occurring at daytime hours (13:00 UT – 22:00 UT) and the lowest values occur at pre-sunrise
        hours (around 8:00 UT). The highest values measured around 11 MHz are observed between 17:00 UT and 20:00 UT from
        September to March evidencing the seasonal trends. The lowest values of around 1.5 MHz occur between 7:00 UT and 9:00
        UT during the June solstice months. Regarding the CCIR prediction shown in Fig. 1(b) and URSI predictions in Fig. 1(c),
        $f_oF_2$ *CCIR* and $f_oF_2$ *URSI*, respectively, it is observed a very similar diurnal and seasonal variation patterns as seen in the

observed values. However, a first look at the $f_oF_2$ *CCIR-RD* in Fig. 1(d) and $f_oF_2$ *URSI-RD* in Fig. 1(e) in the bottom panels reveals
        that the coefficient outputs grossly underestimate/overestimate the $f_oF_2$ in some hours and months as indicated below.
        The $f_oF_2$ *CCIR-RD* in Fig. 1(d) ranges from -20 % (underestimation) to 50 % (overestimation). The higher underestimations are
        observed in September and October from 9:00 UT to 16:00 UT, and later from November to February between 20:00 UT
        and 22:30 UT. There is also an underestimation of 20 % from April to August at around 10:00 UT. On the other hand, the

overestimations are most significant during nighttime hours at almost all months from 23:00 UT to 08:00 UT. The $f_oF_2$ *URSI-RD*
        varies from -15 % to more than 50 %. The most negative deviations are observed only in two small portions of the contour
        plot in Fig. 1(e), which is around 21:00 UT in October, and from 18:00 UT to 22:00 UT in December. However, significant
        positive deviations higher than 50 % are seen around 9:00 UT from March to July, and in the nighttime hours around 23:00
        UT from February to April. From these results, it seems that the URSI (CCIR) sub-routine overestimate (underestimate) $f_oF_2$

more than the CCIR (URSI).
        A more detailed analysis has to be performed to further investigate the level of reliability of each IRI sub-routine. Since the
        data is not significantly drawn from a normally distributed population at the 0.05 % level, the quantitative estimate can be
        achieved by analysing the statistical relationship between IRI- $f_oF_2$ and observed values using the Spearman correlation
        coefficient (*r*). The significance of the calculated *r*-value is examined with a confidence level of 95 % between the hourly

values modelled and observed data. The scatter plots of modelled IRI- $f_oF_2$ using CCIR and URSI coefficients versus the
        observational data are shown in Fig. 2. The results of the calculated *r* are 0.97 for both IRI coefficients. It is shown an almost
        perfect positive correlation.



### 3.2 Performance of IRI-$h_mF_2$

The contour plots of the monthly averaged $h_mF_2$ (in km) observed by the SMK29 and modelled by AMTB, SHU, and BSE
sub-routines and the *RD* (in percent) versus universal time (UT, vertical axis) and month (from September 2017 to August
2018, horizontal axis) are shown in Fig. 3. The color-coded bar on the right-hand side of the upper panels represent $h_mF_2$
ranging from 180 km to 360 km. In the lower panels, the color-coded bar refers to the *RD* and ranges ± 50 % for the three
plots. The observed $h_mF_2$ values in Fig. 3(a) show that the F$_2$-layer is higher during nighttime hours achieving 340 km from
September to December from 1:00 UT to around 03:00 UT. There is also pronounced $h_mF_2$ values between 300 km and 320
km in September equinox and December solstice months from 12:00 UT to 18:00 UT. The daytime average values in March
equinox and June solstice months are usually below 240 km. The pronounced values during nighttime in all months and in
the daytime during September equinox and December solstice months are quite well represented by the AMTB in Fig. 3(b),
SHU in Fig. 3(c), and BSE in Fig. 3(d) as well as the low values during the daytime from March equinox and June solstice
months. Although there are similarities, IRI-2016 predictions have some different aspects as shown by the $h_mF_2$ $_{AMTB-RD}$ in
Fig. 3(e), $h_mF_2$ $_{SHU-RD}$ in Fig. 3(f), and $h_mF_2$ $_{BSE-RD}$ in Fig. 3(g). A visual comparative analysis shows that the SHU agrees
better with the observations since the *RD* encompasses, in general, ±10 % most of the time. The same is not true for AMTB
and BSE predictions.

The $h_mF_2$ $_{AMTB-RD}$ ranges from -10 % to 43 %. The main differences are related to the overestimation of $h_mF_2$ most of the time
in September, October, and from March to August as represented by the hottest color of the palette. It differs especially near
the sunrise period from 7:00 UT to 11:00 UT in the June equinox. The $h_mF_2$ $_{SHU-RD}$ varies from -20 % to 20 %. In general, the
SHU outputs differ only ±10 % from the observation results revealing very good agreement with the observations. Regarding
$h_mF_2$ $_{BSE-RD}$, it ranges from -24 % to 20 %. There are some small periods near sunrise (sunset) that $h_mF_2$ is overestimated
(underestimated), but in general, BSE also represents well the observations. As shown by the results presented in Fig. 3,
SHU and BSE perform better than AMTB in modelling $h_mF_2$. This result is also confirmed by the statistical relationship
through the Spearman $r$ values shown in Fig. 4. Modelling the $h_mF_2$ with the SHU coefficients presents the best scenario with
the $r = 0.86$, as shown in Fig. 4(b). Despite the AMTB in Fig. 4(a) presents the lower correlation ($r = 0.72$), it is important to
note that it is still significant.

### 3.3 Performance of IRI-$f_oE$

The contour plots of the monthly averaged $f_oE$ (in MHz) observed by the SMK29 and modelled by IRI-2016 and the
estimated *RD* (in percent) versus universal time (UT, vertical axis) and month (from September 2017 to August 2018,
horizontal axis) are shown in Fig. 5. The $f_oE$ values are represented by the color-coded bar on the right-hand side, ranging
from 1 MHz to 3.5 MHz for the critical frequency in the upper panels, and ± 20 % for the *RD* in the lower panel.

The observed $f_oE$ in Fig. 5(a) shows a regular diurnal variation, increasing from sunrise to a peak in the afternoon to around
3.5 MHz, and falling until sunset. The low electron density at night makes it difficult to detect the E-region by the Digisonde.
The most intense values around 3.5 MHz are seen during September equinox and December solstice months. The agreement



between IRI-$f_oE$ and observations is very good as shown in Fig. 5(b). The maxima values seen in IRI occur longer than the observations, however. It is shifted two months (April and May) and it starts earlier (July). The $f_oE_{IRI-RD}$ in Fig. 5(c) are positive (overestimation) up to 5 % only, confirming the good IRI-2016 performance in modelling $f_oE$ almost all the time over Santa Maria. There are some considerable differences in a short time in the sunrise and sunset hours. These are critical

periods which may be caused by distortions in the E-region traces due to horizontal gradients in the ionosphere making it difficult to be modelled by IRI, as can be expected by the users. The $r$-value obtained between the modelled and observed values is the highest in this work, showing a very strong positive correlation, as shown in Fig. 6.

**4 Discussion**

The focus of this work is to use the $f_oF_2$, $f_oE$ and $h_mF_2$ measured by the recent Digisonde installed in Santa Maria, Brazil, to

test the performance of the IRI-2016 in the low-latitude ionosphere situated close to the center of the SAMA during the geomagnetically quiet days from September 2017 to August 2018. The results presented in Figs. 1 and 2 show that the $f_oF_2$ predictions obtained with CCIR and URSI coefficients are very similar in a month by month analysis. However, CCIR (URSI) fails underestimating (overestimating) $f_oF_2$ in specific nighttime hours. When the whole period of data is considered, both coefficients gave $r$ equal to 0.97. The correlation is an indication that the model accurately predicts the diurnal and

seasonal trends of $f_oF_2$ over Santa Maria. In general, the IRI user may choose anyone sub-routine to model $f_oF_2$. It is important to stress that the URSI coefficient uses a physical model to obtain $f_oF_2$ over regions not covered by ionosondes, and therefore it is recommended to be used over the oceans. On the other hand, CCIR is recommended to be used over the continents. Note that Santa Maria has located approximately 300 km from the Atlantic Ocean.

The results obtained in this work closely follow the earlier work of Ezquer et al. (2008), who had compared the CCIR and

URSI coefficients with the ionosonde data in Tucumán. They report that, in general, both coefficients give comparable values. However, they also report disagreements among predictions and measurements reaching values of $RD$ close to 50 %. In the Brazilian sector, Batista and Abdu (2004) in a similar comparative study pointed out that the agreements between the URSI values and the observed $f_oF_2$ in São Luís were always better as compared to the CCIR coefficients. They also showed that the $f_oF_2$ after sunset is overestimated for the equatorial station of São Luís. It seems that over the Brazilian territory the

right choice between CCIR and URSI in modelling $f_oF_2$ depends on the location of the users. In the Brazilian equatorial region, CCIR performs better, while in the SAMA region there are no appreciated differences between both. In China, Liu et al. (2019) found that the CCIR performs better than URSI during post-sunset under low solar activity or in the EIA region. For other time and outside the EIA region over China CCIR shows no large difference in performance as compared to URSI. Despite the inclusion of two new model options for the $h_mF_2$ (AMTB and SHU) be an important update in IRI-2016, it is

observed from the results in Figs. 3 and 4 that the AMTB required further improvements. The SHU option performs better over Santa Maria, followed by BSE and the AMTB is worst. The $r$-value of AMTB is 0.72, the lowest observed in the present study. It is even lower than the older BSE coefficient used in the previews versions of the IRI model. Overall, the AMTB (BSE) overestimate (underestimate) the observed values. Therefore, it is recommended the usage of SHU option


when modelling the $h_mF_2$ over Santa Maria. These results agree with the finds of Zhao et al. (2017), who also recommend the
use of SHU option over China region when using IRI-2016 to model $h_mF_2$. Since this is the first evaluation of the three IRI-
$h_mF_2$ options in the Brazilian sector to the author's knowledge, there is no comparison between our work with others
Brazilian equatorial or low latitude regions, and it is suggested as a future study.

Finally, the comparative results presented in Figs. 5 and 6 show that the IRI-predicted $f_oE$ values are in excellent agreement
with observations in Santa Maria. The calculated $r$-value is 0.99. The strong correlation may be explained by the fact that the
E region ionization is subject to solar radiation control, and therefore IRI predicts the E region solar ionization fairly
accurately everywhere in the globe since there is no plasma transport in the E region.

## 5 Conclusions

The present work uses the $f_oF_2$, $f_oE$ and $h_mF_2$ parameters acquired by a recent Digisonde installed in Santa Maria, Brazil,
close to the center of the SAMA, to test the performance of the IRI-2016. Only data collected under quiet conditions from
September 2017 to August 2018 are used to eliminate the effects of geomagnetic disturbances. Monthly average values of
the observed ionospheric parameters are calculated from the daily hourly values and compared with the IRI-2016 predictions
for the same geophysical conditions. The Relative Deviation ($RD$) had been computed using the CCIR and URSI coefficients
to estimate the IRI-$f_oF_2$ performance. The IRI-$h_mF_2$ predictions are evaluated using the $RD$ estimated using the three options
AMTB, SHU, and BSE. The IRI-$f_oE$ performance is also tested. The main findings of the study are as follows:

a)  CCIR and URSI predictions represent the diurnal and seasonal variation patterns of the observed values. $f_oF_2$ $_{CCIR-RD}$
ranges from -20 % (underestimation) to 50 % (overestimation). The higher underestimations are observed in
September and October from 9:00 UT to 16:00 UT, and later from November to February between 20:00 UT and
22:30 UT. There is also an underestimation of 20 % from April to August at around 10:00 UT. The overestimations
are most significant during nighttime hours at almost all months from 23:00 UT to 08:00 UT. The $f_oF_2$ $_{URSI-RD}$ varies
from -15 % to more than 50 %. The most negative deviations are observed at around 21:00 UT in October, and from
18:00 UT to 22:00 UT in December. Significant positive deviations higher than 50 % are seen around 9:00 UT from
March to July, and in the nighttime hours around 23:00 UT from February to April.

    b)  SHU agrees better with the observations than AMTB and BSE for modelling $h_mF_2$. The $h_mF_2$ $_{AMTB-RD}$ ranges from -
10 % to 43 %. The main differences are related to the overestimation of $h_mF_2$ most of the time in September,
October, and from March to August. It differs especially near the sunrise period from 7:00 UT to 11:00 UT in June
equinox. The $h_mF_2$ $_{SHU-RD}$ varies from -20 % to 20 % and, in general, differ only ±10 % from the observation. results
revealing very good agreement with the observations. $h_mF_{2BSE-RD}$ ranges from -24 % to 20 %. There are some small
periods near sunrise (sunset) that $h_mF_2$ is overestimated (underestimated), but in general, BSE also represents well
the observations.

c)  The agreement between IRI-$f_oE$ and observations are very high. However, the maxima values seen in IRI occur
longer than the observations and it is shifted two months (April and May) and it starts earlier (July). The $f_oE_{IRI-RD}$





are positive (overestimation) up to 5 % only, confirming the good IRI-2016 performance in modelling $f_oE$ almost all the time over Santa Maria except in a short time in the sunrise and sunset hours.

As a general conclusion of this work, it is shown that both CCIR and URSI coefficients have high accuracy in predicting $f_oF_2$
over Santa Maria. The same is true for IRI-$f_oE$. However, it is recommended the users to use the SHU coefficient as the first option to modelling $h_mF_2$ over Santa Maria, which is different from the recommendation of IRI-2016.

**Author contribution**

J. Moro conceived the study, processed the DPS data, designed the data analysis and leaded writing this manuscript.

J. Xu assisted to review the manuscript and discuss the results of the study.

C. M. De Nardin assisted to review the manuscript and discuss the results of the study.

L. C. A. Resende assisted to process the DPS data analysis and discuss the results of the study.

R. P. Silva assisted to review the manuscript and discuss the results of the study.

S. S. Chen assisted to review the manuscript and discuss the results of the study.

G. A. S. Picanço assisted to review the manuscript and discuss the results of the study.

L. Zhengkuan assisted to review the manuscript and discuss the results of the study.

H. Li assisted to review the manuscript and discuss the results of the study.

C. Yan assisted to review the manuscript and discuss the results of the study.

C. Wang assisted to review the manuscript and discuss the results of the study.

N. J. Schuch assisted to review the manuscript and discuss the results of the study.

**Competing interests**

The authors declare that they have no conflict of interest

**Acknowledgments**

J. Moro would like to acknowledge the China-Brazil Joint Laboratory for Space Weather (CBJLSW), National Space Science Center (NSSC), Chinese Academy of Sciences (CAS) for supporting his Postdoctoral fellowship, and the National
Council for Scientific and Technological Development (CNPq) for the grant 429517/2018-01. C. M. Denardini thanks CNPq for the fellowship under the number 303643/2017-0. L. C. A. Resende would like to acknowledge the CBJLSW for supporting her Postdoctoral fellowship. R. P. Silva acknowledges the support from CNPq for the grant number 300329/2019-9. S. S. Chen thanks the Coordination for the Improvement of Higher Education Personnel (CAPES) for supporting his Ph.D. (grand number 88887.362982/2019-00). G. A. S. Picanço thanks CAPES (grant 88887.351778/2019-
00. N. J. Schuch thanks CNPq for the fellowship under the number 300886/2016-0. The authors also thank the OMNIWEB (https://omniweb.gsfc.nasa.gov/form/dx1.html) for providing $F_{10.7}$ index, sunspot number, and the $Kp$ index used in the classification of the days. The Digisonde data from Santa Maria can be downloaded upon registration at the Embrace





webpage from INPE Space Weather Program in the following link: http://www2.inpe.br/climaespacial/portal/en/. The authors acknowledge the support of the Federal University of Santa Maria (UFSM) Central Administration.

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



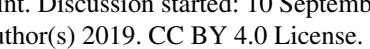


**Table 1.** 27-day averaged values of F10.7, sunspot number, and the number of quiet-days from September 2017 to August 2018 considered in this work.

| Year | Month | $F_{10.7}$ | Sunspot number | Amount of quiet days |
|---|---|---|---|---|
| 2016 | September | 80.1 | 18 | 11 |
| | October | 72.1 | 8 | 18 |
| | November | 70.3 | 8 | 17 |
| | December | 69.4 | 9 | 21 |
| 2017 | January | 67.7 | 6 | 24 |
| | February | 70.4 | 12 | 20 |
| | March | 67.6 | 1 | 21 |
| | April | 70.3 | 10 | 24 |
| | May | 70.7 | 7 | 22 |
| | Jun | 74.5 | 16 | 23 |
| | July | 74.7 | 12 | 26 |
| | August | 71.2 | 1 | 21 |
| | | | Total | 248 |




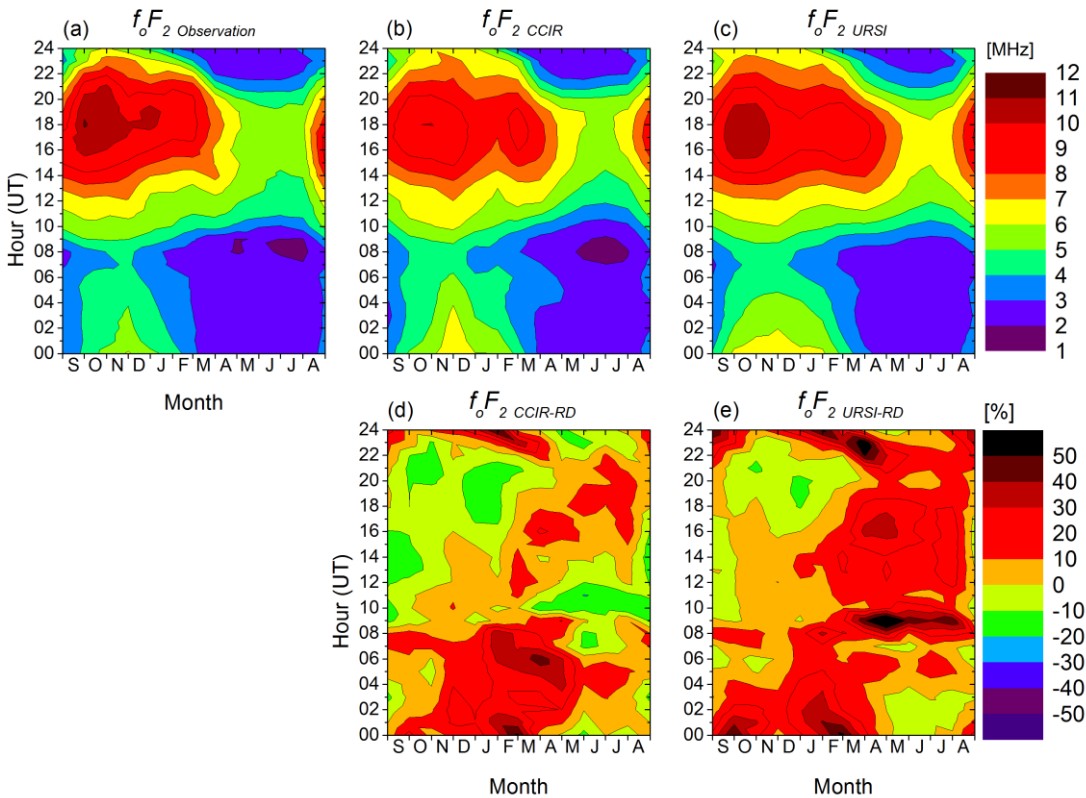

**Fig.1:** Contour plot of the monthly averaged $f_oF_2$ (a) measured by the Santa Maria Digisonde (SMK29), provided by (b) CCIR and (c) URSI coefficients. The respective Relative Deviation (*RD*) in percent is placed in the lower panels for (d) CCIR and (e) URSI.






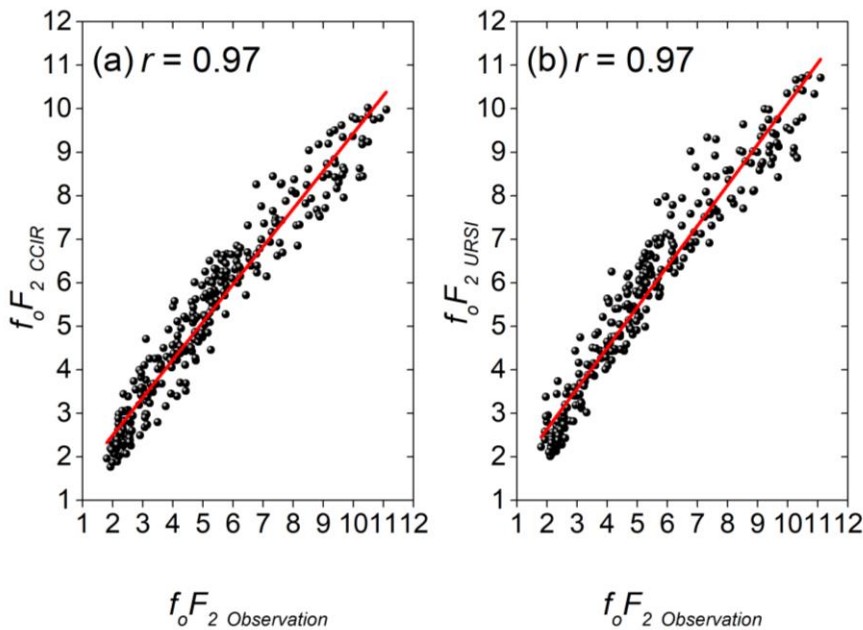

**Fig. 2:** Scatter plots depicting the comparison between the (a) $f_oF_2$ $_{CCIR}$ and (b) $f_oF_2$ $_{URSI}$ versus $f_oF_2$ $_{Observation}$.

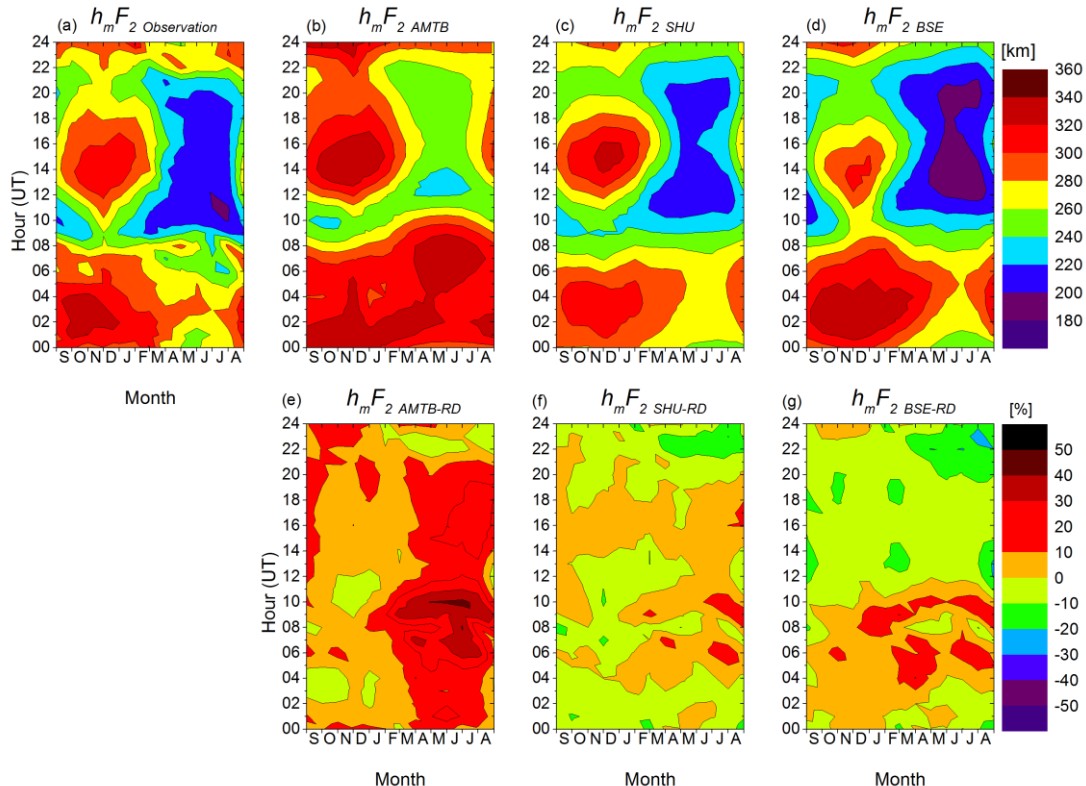

**Fig. 3:** Contour plot of the monthly averaged $h_mF_2$ (a) measured by the Santa Maria Digisonde (SMK29), provided by (b)
AMTB, (c) SHU, and (d) BSE coefficients. The respective Relative Deviation (*RD*) in percent is placed in the lower panels
for (e) AMTB, (f) SHU, and (g) BSE.





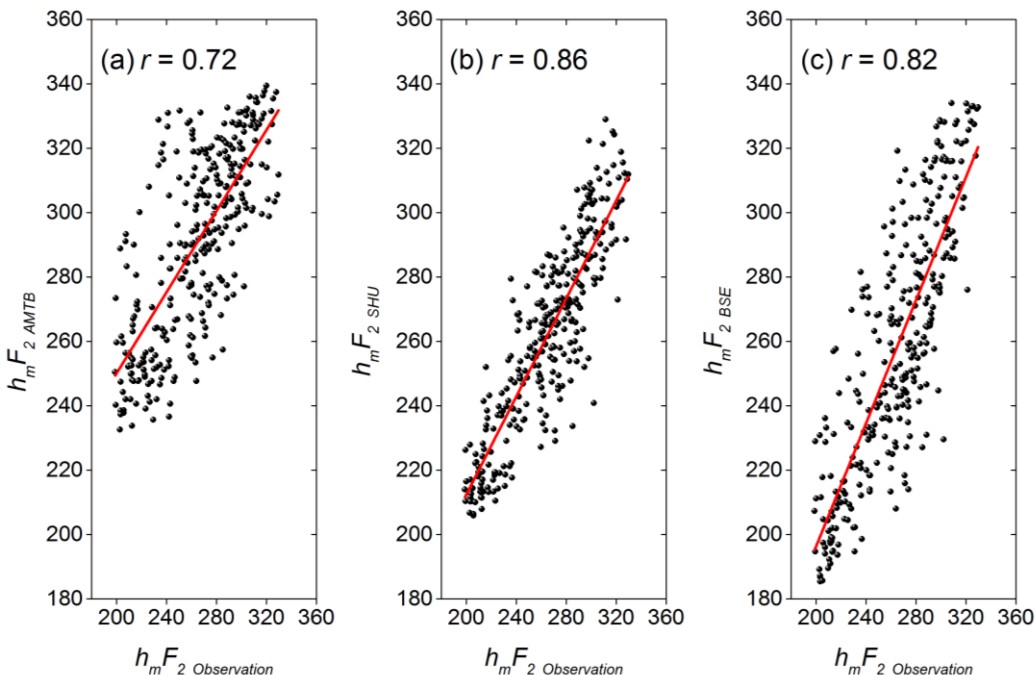

**Fig. 4:** Scatter plots depicting the comparison between the (a) $h_mF_{2\ AMTB}$, (b) $h_mF_{2\ SHU}$ and (c) $h_mF_{2\ BSE}$ versus
$h_mF_{2\ Observation}$.





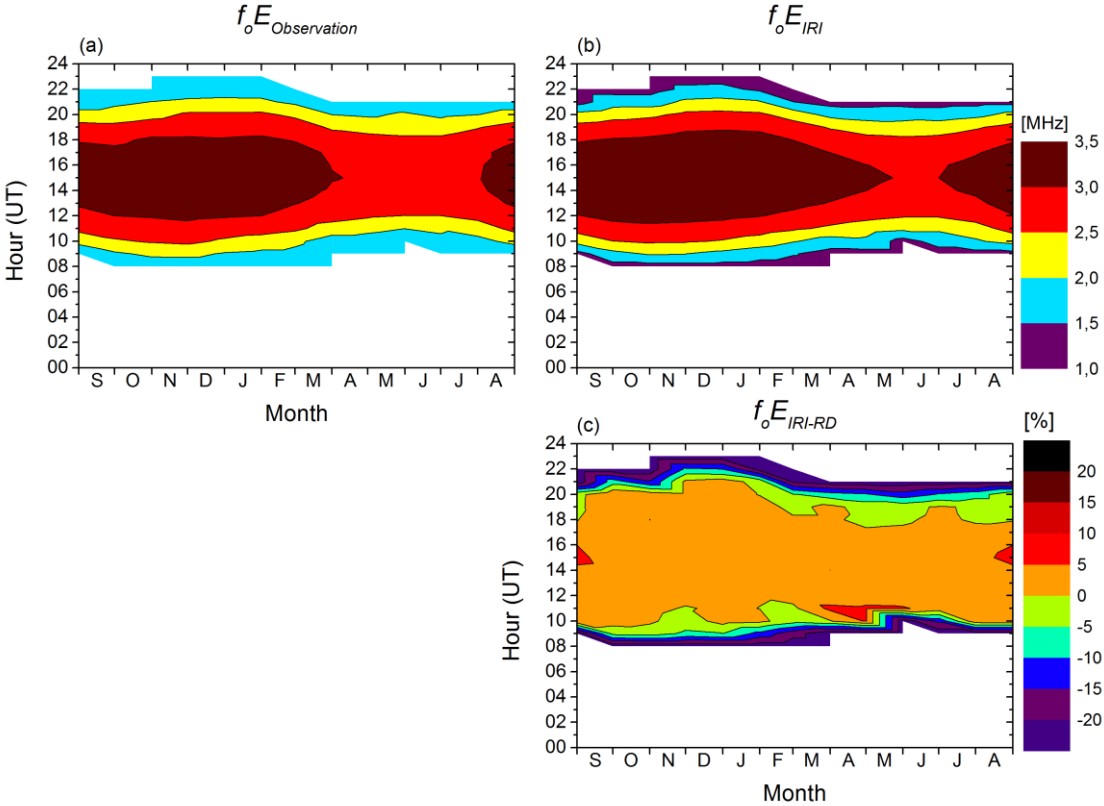

**Figure 5.** Contour plot of the monthly averaged $f_oE$ (a) measured by the Santa Maria Digisonde (SMK29), provided by (b) IRI, and (c) the Relative Deviation (*RD*) in percent.






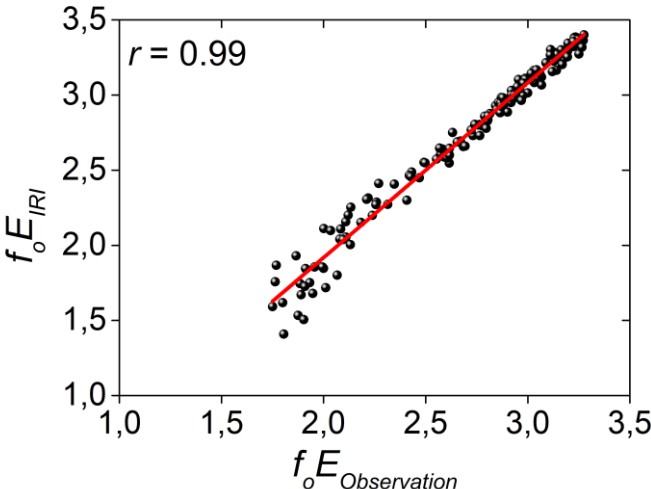

**Fig. 6:** Scatter plot depicting the comparison between the $f_oE_{IRI}$ versus $f_oE_{\,Observation}$.