# Peer review of "Performance of the IRI-2016 over Santa Maria, a Brazilian low-latitude station located in the central region of the South American Magnetic Anomaly (SAMA)"

_Annales Geophysicae, 2019_

## Referee Comment (RC1) · Anonymous Referee #1 · 7 Oct 2019

Review comments on paper "Performance of the IRI-2016 at the Brazilian low-latitude ionosphere over the South America Magnetic Anomaly during solar minimum " by Moro et al., 2019

The short paper is devoted to validating the IRI 2016 model with ionosonde data collected over a single location, Santa Maria, in Brazil. Results are presented for three ionospheric parameters, foE, foF2 and hmF2. Studies of this nature are important in showing the performance of climatological models such as the IRI; and subsequently assist in their improvement. Data from Santa Maria will be very vital for the updating and improving the IRI model and I suggest that the authors/owners of this ionosonde

make the data available to the scientific community. In this regard, I would like to suggest that the authors indicate where the data can be accessed or whether it is being sent to the GIRO database that collects data for all digisondes all over the world. Having said this, I think, the title should have been specific that the study is done over one location, aside from this, the reader may think that this is a regional analysis focusing on the entire Brazilian low-latitude.

The authors presented the performance of the IRI model using different hmF2 options and came up with a conclusion that the Shubin option estimates hmF2 better than the other options over SMK29. This is consistent with some studies carried out over other regions such as China.

Statistical analyses was based on correlation coefficients and relative deviation (RD). In their discussion, the authors did not indicate quantitatively how their values compare with other studies at similar latitudes. I understand that their location is near SAMA. However, they should have compared their statistical values to other studies in other low latitude regions. Also other previous studies may be using different statistical parameters such as root mean square error which is simple to calculate for the sake of benchmarking the authors results with existing studies.

In the entire paper, there are a number of language usage errors that should be corrected.

Below are some comments that may be helpful:

1. The authors used the quiet time data in the analysis and the criterion used was "summation of Kp values less or equal to 24" . I wondered whether this had some references. Otherwise, wouldn't it be straightforward to for-example simply use Kp<=2? 2. In line 75, there is a spelling error in "angle=-37 degrees" 3. Line 140, the highest values during day-time hours between 13:00 UT -22:00 UT. According to the authors statement in the paper, this corresponds to 10:00 LT – 19:00 LT. It appears that the high values at 19:00 LT may be related to pre-reversal enhancement, and not necessarily

to day-time hours? Please cross-check and correct where necessary. 4. In line 170 and elsewhere in the paper: The authors state "There is also pronounced hmF2 values between 300 km and 320 km ...". I think this should be reworded. "pronounced" in what context? 5. Subsections 3.1 and 3.2 can be combined and discussed at once as they deal with F2-region parameters. This will eliminate some repetitions in the narrative. 6. There seems to be a mistake in "Years" in Table 1. The Years indicated are 2016 and 2017; and yet the authors state that their study was conducted during 2017-2018?

---

## Author Comment (AC1) · 9 Jan 2020

**RESPONSE LETTER TO REVIEWER #1**

*We thank the reviewer for taking the time to review our manuscript and provide important comments and suggestions. Please, see below our responses:*

1) The short paper is devoted to validating the IRI 2016 model with ionosonde data collected over a single location, Santa Maria, in Brazil. Results are presented for three ionospheric parameters, foE, foF2 and hmF2. Studies of this nature are important in showing the performance of climatological models such as the IRI; and subsequently assist in their improvement. Data from Santa Maria will be very vital for the updating and improving the IRI model and I suggest that the authors/owners of this ionosonde make the data available to the scientific community. In this regard, I would like to suggest that the authors indicate where the data can be accessed or whether it is being sent to the GIRO database that collects data for all digisondes all over the world. Having said this, I think, the title should have been specific that the study is done over one location, aside from this, the reader may think that this is a regional analysis focusing on the entire Brazilian low-latitude.

*Our response: We thank the reviewer for the interest in the Digisonde data collected in Santa Maria. The data are available to the scientific community. For access the numerical data in the DIDBase, the users have to install the SAOEXPLORER and create an account. Alternatively, the users may download the data upon registration at the Embrace webpage from INPE Space Weather Program in the following link: http://www2.inpe.br/climaespacial/portal/en/ as indicated in the Acknowledgements. Regarding the title of our work, we agree with the Reviewer #1 and changed the title. In the present version the tile is "Performance of the IRI-2016 over Santa Maria, a Brazilian low-latitude station located in the central region of the SAMA".*

(2) The authors presented the performance of the IRI model using different hmF2 options and came up with a conclusion that the Shubin option estimates hmF2 better than the other options over SMK29. This is consistent with some studies carried out over other regions such as China. Statistical analyses was based on correlation coefficients and relative deviation (RD). In their discussion, the authors did not indicate quantitatively how their values compare with other studies at similar latitudes. I understand that their location is near SAMA. However, they should have compared their statistical values to other studies in other low latitude regions. Also other previous studies may be using different statistical parameters such as root mean square error which is simple to calculate for the sake of benchmarking the authors results with existing studies.

*Our response: We thank the Reviewer #1 for these suggestions. We did not indicate quantitatively how our values are compared with other studies at similar latitudes because we understand that this comparison is not fair. The main reason is due to the absence of a geomagnetic anomaly in other regions (as mentioned by the Reviewer). In addition, other regions may have the features of the Equatorial Ionization Anomaly (EIA). Such characteristics impose large variabilities that there are no present over Santa Maria. Therefore, the best we can do is to compare our values qualitatively, as we did with the works Batista and Abdu (2004), Ezquer et al. (2008), and Liu et al. (2019), among others. Regarding the method, there are several approaches that we could follow to compare our results with other studies. We chose the correlation coefficients and the relative deviation, which we notice that are the most used by the scientific community.*

In the entire paper, there are a number of language usage errors that should be corrected. Below are some comments that may be helpful:

1. The authors used the quiet time data in the analysis and the criterion used was "summation of Kp values less or equal to 24". I wondered whether this had some references. Otherwise, wouldn't it be straightforward to for-example simply use Kp<=2?

*Our response: We usually classify the quiet days considering that the three-hourly Kp value never exceeds a value of 3 over the entire day. Please, see below some references:*

*Moro et al. (2012), Ann. Geophys., 30, 1159–1168, doi:10.5194/angeo-30-1159-2012.*
*Denardini et al. (2018), Radio Science, 53, 288-302, doi: 10.1002/2017RS006477.*
*Denardini et al. (2018), Radio Science, 53, 379–393, doi: 10.1002/2018RS006540.*

2. In line 75, there is a spelling error in "angle=-37 degrees"

*Our response: We corrected it.*

3. Line 140, the highest values during day-time hours between 13:00 UT -22:00 UT. According to the authors statement in the paper, this corresponds to 10:00 LT – 19:00 LT. It appears that the high values at 19:00 LT may be related to pre-reversal enhancement, and not necessarily to day-time hours?

*Our response: We believe that these highest values are not related to the pre-reversal enhancement since Santa Maria is located at -30º. The previous studies over the Brazilian region (Abdu et al., 1981, doi:10.1029/JA086iA08p06836; Batista et al., 1986, J Geophys Res 91:12055–12064) have shown that the intensity of the pre-reversal enhancement is significant in the equatorial region. However, as Santa Maria is a transition region (low/mid-latitude), we attribute these highest values to the daytime dynamics. In the present version of the manuscript, we added this information.*

4. In line 170 and elsewhere in the paper: The authors state "There is also pronounced hmF2 values between 300 km and 320 km ...". I think this should be reworded. "pronounced" in what context?

*Our response: The context is 'more intense' values. We changed the word 'pronounced' by 'more intense' in lines 171 and 173.*

5. Subsections 3.1 and 3.2 can be combined and discussed at once as they deal with F2-region parameters. This will eliminate some repetitions in the narrative.

*Our response: We thank the reviewer for this suggestion. We combined the Subsections 3.1 and 3.2 and it is '3.1 Performance of IRI in the $F_2$-region parameters' in the present version of the manuscript.*

6. There seems to be a mistake in "Years" in Table 1. The Years indicated are 2016 and 2017; and yet the authors state that their study was conducted during 2017-2018?

*Our response: Apologies for our mistake in Table 1. We correct it to the year 2017-2018.*

*Finally, we would like to thank again the Reviewer #1 for he/she assistance in evaluating the paper, and the comments for improvements.*

---

## Referee Comment (RC2) · Anonymous Referee #2 · 24 Jan 2020

Review of the article "Performance of the IRI-2016 at the Brazilian low-latitude ionosphere over the South America Magnetic Anomaly during solar minimum." 2019 Moro et. al.

In this article, the authors perform an analysis of foF2, hmF2, and foE between measurements made by a digisonde in Santa Maria, Brazil, and the output of various IRI-2016 model sub-routines. The article provides a unique analysis on the performance and validation of IRI-2016 in the South American Magnetic Anomaly region. The authors use the relative deviation, an approach used in similar studies, and the Spearman correlation coefficient to provide a quantitative analysis of model performance. Based

on the results, the authors give recommendations on which IRI-2016 sub-routines to use.

The data used in their analysis includes ionograms made during geomagnetically quiet times. The authors use the Kp index to identify periods of quiet time by making sure that the sum of the eight 3-hour Kp indices for the day is less than 24. For clarification, are the authors using the eight 3-hour Kp indices made prior to a given measurement, or is it just based on the eight 3-hour Kp indices for a given universal time day? If it is the latter case, it would seem that determining whether a measurement occurred during geomagnetically quiet periods would depend on Kp values for times occurring after the measurement. This may falsely categorize some measurements.

In their analysis of foF2, the authors find that both the CCIR and URSI sub-routines provide correlation coefficients of r=.97, and suggest that the user may use either sub-routine to model foF2 over the Santa Maria region. However, the sentence starting on line 210 suggests that users should use URSI over the oceans and CCIR over the continents. This statement seems too general and beyond the scope of the paper. The statement also seems to be contradicted by the findings of Batista and Abdu (2004) who found URSI outperformed CCIR over São Luís in the Brazilian sector. Also, the authors should make sure that the Batista and Abdu (2004) reference is including in their bibliography. I could not locate it.

In the discussion section, the authors provide some comparison with results from previous studies. For instance, the finding that URSI and CCIR provide comparable results was also found to be true in Ezquer et al. (2008). Additionally, Zhao et al. (2017) also found that SHU outperformed BSE and AMTB in hmF2 predictions in the China region.

The authors find the highest correlation between SMK29 and IRI-2016 for foE values and suggest that a potential reason for this is because the E-region ionization is controlled primarily by solar radiation and IRI can predict this radiation fairly accurately around the globe. The author make it clear that this is simply a proposed explanation,

however, this claim could be substantiated by providing comparisons with other studies. To the authors knowledge, do previous studies exist showing the similarly high performance of IRI-2016 foE predictions for other parts of the globe?

Below are some corrections to grammatical errors and suggestions for potential changes to the text to help improve readability:

Line 38: "measurements as the worldwide" -> "measurements such as the worldwide"

Line 41: "were released by the IRI Working Group since the 1970s in order to constantly revising the model to remain it up to date and accurate as possible" -> "have been released by the IRI Working Group since the 1970s in order to update the model to keep it as accurate as possible"

Line 52: "E heights" -> "E-region heights"

Line 77: "dip anlge" -> "dip angle"

Line 83: "dip = -49.8" -> "dip angle = -49.8"

Line 111: "the comparison is made considering the currently three options for determining IRI-hmF2" ->" the comparison is made using the three currently available options for determining IRI-hmF2"

Line 114: "deduced of ionograms" -> "deduced from ionograms"

Line 213: "Santa Maria has located" -> "Santa Maria is located"

Line 224: "Despite the inclusion of two new model options for the hmF2 (AMTB and SHU) be an important update" – I don't have a suggestion but this sentence should be reworded.

Line 265: "it is recommended the users to use" -> "it is recommended that the users use"

---

## Author Comment (AC2) · 29 Jan 2020

**RESPONSE LETTER TO REVIEWER #2**

*We thank the reviewer for taking the time to review our manuscript and provide important comments and suggestions. Please, see below our responses:*

1) In this article, the authors perform an analysis of foF2, hmF2, and foE between measurements made by a digisonde in Santa Maria, Brazil, and the output of various IRI-2016 model sub-routines. The article provides a unique analysis on the performance and validation of IRI-2016 in the South American Magnetic Anomaly region. The authors use the relative deviation, an approach used in similar studies, and the Spearman correlation coefficient to provide a quantitative analysis of model performance. Based on the results, the authors give recommendations on which IRI-2016 sub-routines to use.

The data used in their analysis includes ionograms made during geomagnetically quiet times. The authors use the Kp index to identify periods of quiet time by making sure that the sum of the eight 3-hour Kp indices for the day is less than 24. For clarification, are the authors using the eight 3-hour Kp indices made prior to a given measurement, or is it just based on the eight 3-hour Kp indices for a given universal time day? If it is the latter case, it would seem that determining whether a measurement occurred during geomagnetically quiet periods would depend on Kp values for times occurring after the measurement. This may falsely categorize some measurements.
*Our response: We used the eight 3-hours Kp made prior to the measurements.*

In their analysis of foF2, the authors find that both the CCIR and URSI sub-routines provide correlation coefficients of r = .97, and suggest that the user may use either sub-routine to model foF2 over the Santa Maria region. However, the sentence starting on line 210 suggests that users should use URSI over the oceans and CCIR over the continents. This statement seems too general and beyond the scope of the paper. The statement also seems to be contradicted by the findings of Batista and Abdu (2004) who found URSI outperformed CCIR over São Luís in the Brazilian sector. Also, the authors should make sure that the Batista and Abdu (2004) reference is including in their bibliography. I could not locate it.
*Our response: We absolutely agree with the reviewer that the statement 'the users should use URSI over the oceans and CCIR over the continents' is too general and beyond the scope of the paper. Indeed, reading the paragraph repeatedly we see now that the last 3 sentences of the paragraph do not add relevant information and might confuse the readers. Therefore, we deleted these sentences in the new version of the manuscript. Regarding the results of Batista and Abdu (2004), we explained in lines 219-223 that over the Brazilian territory the right choice between CCIR and URSI in modeling foF2 depends on the location of the users (equatorial, midlatitudes). The reference Batista and Abdu (2004) was not included in the bibliography because of lack of attention. We added it in this last version of the manuscript.*

In the discussion section, the authors provide some comparison with results from previous studies. For instance, the finding that URSI and CCIR provide comparable results was also found to be true in Ezquer et al. (2008). Additionally, Zhao et al. (2017) also found that SHU outperformed BSE and AMTB in hmF2 predictions in the China region.

The authors find the highest correlation between SMK29 and IRI-2016 for foE values and suggest that a potential reason for this is because the E-region ionization is controlled primarily by solar radiation and IRI can predict this radiation fairly accurately around the globe. The author make it clear that this is simply a proposed explanation, however, this claim could be substantiated by

providing comparisons with other studies. To the authors knowledge, do previous studies exist showing the similarly high performance of IRI-2016 foE predictions for other parts of the globe?

***Our response:*** *As far as we know there is no similar study comparing NmE (or foE) obtained from IRI with Digisonde data. However, there has been a significant amount of studies about the E-region modelling, most of them based on photochemical approximation. We did not include these references in this work because they are beyond the scope of our paper.*

Below are some corrections to grammatical errors and suggestions for potential changes to the text to help improve readability:

***Our response:*** *We acknowledge the corrections/ suggestions given by the reviewer. We corrected them accordingly.*

Line 38: "measurements as the worldwide" -> "measurements such as the worldwide" *Done*
Line 41: "were released by the IRI Working Group since the 1970s in order to constantly revising the model to remain it up to date and accurate as possible" -> "have been released by the IRI Working Group since the 1970s in order to update the model to keep it as accurate as possible" *Done*
Line 52: "E heights" -> "E-region heights" *Done*
Line 77: "dip anlge" -> "dip angle" *Done*
Line 83: "dip = -49.8" -> "dip angle = -49.8" *Done*
Line111: "the comparison is made considering the currently three options for determining IRI-hmF2" ->" the comparison is made using the three currently available options for determining IRI-hmF2" *Done*
Line 114: "deduced of ionograms" -> "deduced from ionograms" *Done*
Line 213: "Santa Maria has located" -> "Santa Maria is located" *This sentence was deleted*
Line 224: "Despite the inclusion of two new model options for the hmF2 (AMTB and SHU) be an important update" – I don't have a suggestion but this sentence should be reworded. *We have reworded*
Line 265: "it is recommended the users to use" -> "it is recommended that the users use" *Done*

.

*Finally, we would like to thank again the Reviewer #2 for he/she assistance in evaluating the paper, and the comments for improvements.*

---

## Author Response (AR2)

**RESPONSE LETTER TO REVIEWER #1**

*We thank the Reviewer #1 and the Topical Editor for taking the time to review our manuscript again. We have corrected the minor accordingly. Please, see Lines 144-145.*